# Injecting a Structural Inductive Bias into a Seq2Seq Model by Simulation

## Abstract

Strong inductive biases enable learning from little data and help generalization outside of the training distribution. Popular neural architectures such as Transformers lack strong structural inductive biases for seq2seq NLP tasks on their own. Consequently, they struggle with systematic generalization beyond the training distribution, e.g. with extrapolating to longer inputs, even when pre-trained on large amounts of text. We show how a structural inductive bias can be injected into a seq2seq model by pre-training it to simulate structural transformations on synthetic data. Specifically, we inject an inductive bias towards Finite State Transducers (FSTs) into a Transformer by pre-training it to simulate FSTs given their descriptions. Our experiments show that our method imparts the desired inductive bias, resulting in improved systematic generalization and better few-shot learning for FST-like tasks.

## 1 Introduction

Inductive biases, i.e. the preferences and the abstract knowledge a model brings to the task, enable a model to learn from small amounts of data and generalize systematically outside of the training distribution. While seq2seq models perform very well on in-distribution data on many NLP tasks, they usually lack structural inductive biases and consequently struggle with systematic generalization. Previous work has shown that this includes generalization to unseen combinations of known substrings (Lake & Baroni, 2018; Keysers et al., 2020), extrapolation to longer inputs (Hupkes et al., 2020) and deeper recursion (Kim & Linzen, 2020).

Integrating structural inductive biases into seq2seq models is challenging. One popular approach is to develop specialized architectures (Zheng & Lapata, 2021; Kim, 2021; Lindemann et al., 2023), which makes it difficult to precisely control and adjust the nature of the inductive bias as the architecture would need to be changed and models re-trained. Recently, some works instead have tried to inject inductive biases into seq2seq models by means of pre-training on a well-chosen synthetic task (Krishna et al., 2021; Wu et al., 2021; 2022) or meta-learning on a distribution of synthetic tasks (McCoy et al., 2020; McCoy & Griffiths, 2023) using MAML (Finn et al., 2017). Here, the inductive bias can be controlled by the choice of the synthetic task. However, meta-learning with MAML scales poorly because it requires expensive second-order derivatives and standard pre-training can be less effective (McCoy & Griffiths, 2023).

In this work, we present a computationally inexpensive way of injecting a structural inductive bias into a Transformer. We focus specifically on introducing an inductive bias that is helpful for tasks that traditionally have been approached with Finite State Transducers (FSTs). We choose FSTs because they are formally well understood, are easy to generate automatically, and are one of the simplest computational devices that are useful in NLP applications. While we focus on FSTs, the methodology is fairly general and our approach also provides a starting point for incorporating more general structural biases, provided by more expressive formalisms such as Pushdown Transducers.

Our approach (SIP, for **S**imulation-**I**nduced **P**rior) is simple (see Fig. 1): given a representation of an FST and an input string, a Transformer is pre-trained to predict what the output of the FST is on the given input. We assume that FSTs are not specified for fine-tuning on downstream tasks, so we replace the FST with tunable embeddings and fine-tune the model solely on input/output examples. We show that SIP improves accuracy on systematic generalization and few-shot learning for 'FST-like' downstream tasks, demonstrating that the desired inductive bias has been imparted. SIP not only improves systematic generalization on FST tasks similar to those seen during pre-training but

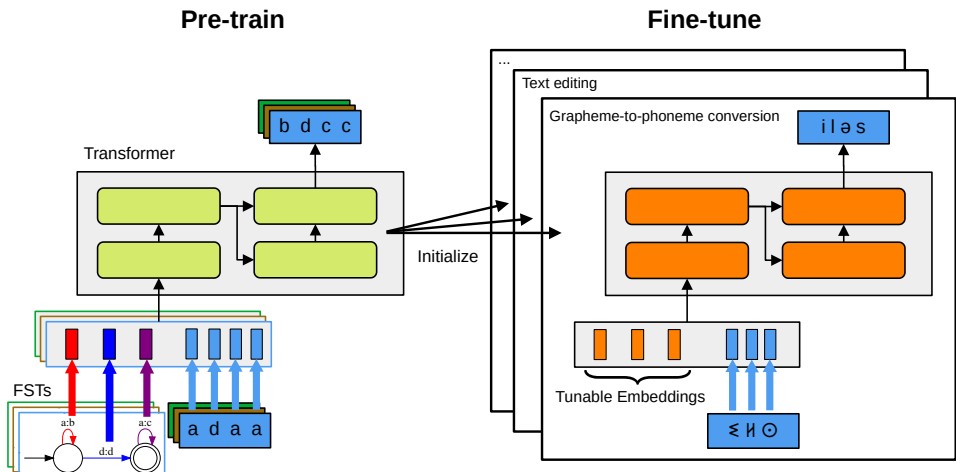

Figure 1: Left: Pre-training a Transformer to simulate automatically generated FSTs. Right: fine-tuning the Transformer and the prefix where the FST used to be on a downstream task by using only input/output pairs. Tunable parameters are represented in orange.

also on ones that are structurally more complex. The same pre-trained model achieves strong results on few-shot learning on text editing (e.g. Jane Doe → J. Doe) and grapheme-to-phoneme conversion, which traditionally have been approached with FSTs. Our contributions are:

- a simple, adjustable and efficient method to inject a structural inductive bias for FST-like tasks into a Transformer.
- better systematic generalization on tasks beyond the pre-training distribution.
- strong results when transferring to natural FST-like data, as demonstrated on low-resource grapheme-to-morpheme conversion.

## 2 RELATED WORK

**Systematic generalization.** Systematic generalization refers to the ability of a model to generalize (or extrapolate) beyond its training distribution in a systematic way that aligns with how humans generalize. Systematic generalization has been shown to be difficult for standard seq2seq models in contexts such as semantic parsing (Finegan-Dollak et al., 2018), machine translation (Li et al., 2021) and algorithmic reasoning (Deletang et al., 2023), in particular to unseen combinations of sub-strings, longer inputs as well as deeper recursion (Keysers et al., 2020; Kim & Linzen, 2020).

A range of approaches have been developed to tackle this, with many works focusing on specialized architectures (Guo et al., 2020; Zheng & Lapata, 2021; Kim, 2021; Lindemann et al., 2023). Furrer et al. (2020) find that the specialized architectures they consider do not transfer well to tasks beyond the context in which they were designed. This highlights the importance of being able to adjust inductive biases more easily than re-designing the architecture of a model. Large-scale pretraining on natural language has been widely successful in NLP (e.g. for few-shot learning) and has also been shown to help with systematic generalization (Furrer et al., 2020). However, challenges remain even for LLMs such as GPT-3 and PALM (Qiu et al., 2022; Dziri et al., 2023). The methodology we present in this work can be used to create additional material for LLM pre-training. Here we focus on smaller models and leave this to future work.

**Pre-training with synthetic tasks.** Pre-training a model on a synthetic task to introduce specific inductive biases has been explored by several recent works. Krishna et al. (2021) identify useful 'skills' for news summarization and develop a pre-training task accordingly. LIME (Wu et al., 2021) targets mathematical reasoning and is pre-trained on symbolic string manipulation that resembles deductive, abductive and inductive reasoning. Wu et al. (2022) investigate a range of simple synthetic tasks for pre-training and show that some perform remarkably well across a range of downstream tasks. Papadimitriou & Jurafsky (2023) consider several synthetic languages to investigate which helps most as pre-training data for language modelling on English. In contrast to these works,

our approach targets simulating a computational device and maintains a closer connection to the pre-training setting because of the tunable prefix.

A challenge for using individually hand-crafted tasks is to cover a sufficient space of phenomena that are relevant to downstream tasks. Instead of training on a single task only, McCoy et al. (2020); McCoy & Griffiths (2023) meta-learn on a distribution of tasks using MAML (Finn et al., 2017). They show that this can be helpful for low-resource language modelling on simple English utterances (McCoy & Griffiths, 2023). Our approach also uses a distribution of tasks but it scales better than MAML-based methods because MAML requires computing and storing second-order derivatives. For example, the Transformer we train has a magnitude more parameters than the LSTM of McCoy & Griffiths (2023) and can be pre-trained on a smaller GPU (A100 vs RTX 2080 TI). In addition, as the complexity of each individual task grows, MAML requires more examples per task. We circumvent this by using a compact and unambiguous description of each task instead.

**Simulating execution.** The idea of using a neural network to predict the outcome of the execution of a computational device or code has come up in several contexts over the last few years. Early work by Zaremba & Sutskever (2014) investigates it as a challenging benchmark for LSTM-based seq2seq models. Recent works have explored simulating (aspects) of code execution for various down-stream applications, such as program synthesis (Austin et al., 2021), debugging and code analysis (Bieber et al., 2022) as well as reverse engineering (Pei et al., 2021). Closer to our setup, Finlayson et al. (2022) train a Transformer to interpret regular expressions: given a regular expression and a string, the task is to decide if the string is in the regular language. There are two crucial differences between their work and ours: (i) they investigate the empirical capabilities of Transformers to simulate regular expressions while we use simulation to introduce structural inductive biases for downstream tasks, and (ii) they consider binary outputs whereas we consider sequential outputs.

## 3 FINITE STATE TRANSDUCERS

We briefly review Finite State Transducers (FSTs) which we use in our experiments. FSTs are closely related to Finite State Automata (FSAs). While an FSA describes a set of strings, an FST describes a *relation* between strings, i.e. a set of pairs $(x, y)$, where $x$ is an input $y$ is an output.

FSTs can be visualized as labelled directed graphs (see Fig. 2), where the nodes are called *states* and the edges are called *transitions*. Consider the path $\texttt{q0} \xrightarrow{\texttt{a:b}} \texttt{q1} \xrightarrow{\texttt{a:b}} \texttt{q1} \xrightarrow{\texttt{b:b}} \texttt{q2}$ in Fig. 2b. This path is called an *accepting path* because it starts in an *initial* state (indicated by an arrow 'from nowhere' pointing to the state), and it ends in a *final* state (indicated by double circles). An accepting path shows what an input can be mapped to. In this case, the path shows that the FST transduces the input $\texttt{aab}$ into the output $\texttt{bbb}$. We can read off which input an accepting path associates an output to by concatenating all the strings along the path occurring before ':'. The output can be determined by concatenating the strings after ':'. Hence, each transition $\xrightarrow{\sigma : \rho}$ can be thought of as 'replacing' $\sigma$ by $\rho$. Inserting and deleting can be achieved by means of the empty string, written as $\epsilon$. For example, Fig. 2a 'replaces' every second $\texttt{a}$ by an empty string, effectively deleting them.

In general, an input can be paired with arbitrarily many different outputs. We call an FST $f$ **functional** if every input $x$ is paired with at most one output $y$, and use the notation $f(x)$ to refer to $y$. All FSTs we consider here are functional. We also use set notation on FSTs, e.g. if $f_1$ and $f_2$ are FSTs expressing relations $R_1$ and $R_2$, we refer to the FST expressing $R_1 \cup R_2$ as $f_1 \cup f_2$.

In this work, we focus mainly on **deterministic** FSTs, which are a less expressive sub-class of the functional FSTs that are particularly easy to generate automatically. We will use deterministic and non-deterministic FSTs to investigate generalization across different sub-classes of FSTs. An FST is called deterministic if (i) it has a unique initial state, (ii) for all states $q$ and input symbols $\sigma$ there is at most one transition $q \xrightarrow{\sigma : \rho} q'$ and (iii) $\sigma \neq \epsilon$. Intuitively, this means that in any state, for an input symbol $\sigma$ there is at most one possible next state and one possible output, and hence for any input string there is at most one path that is compatible with it. Because of this, we can always determine a prefix of the output string by looking only at a *prefix* of the input string and ignoring the rest. For example, consider the input prefix $\texttt{aa}$. In the deterministic FST in Fig. 2a, we know that the output has to start with $\texttt{a}$ because there is only one path that is compatible with $\texttt{aa}$. In contrast, in the non-deterministic FST in Fig. 2b, there are two paths that are compatible with $\texttt{aa}$ that have different outputs. In that case, we can only determine the output once we look at the last symbol of

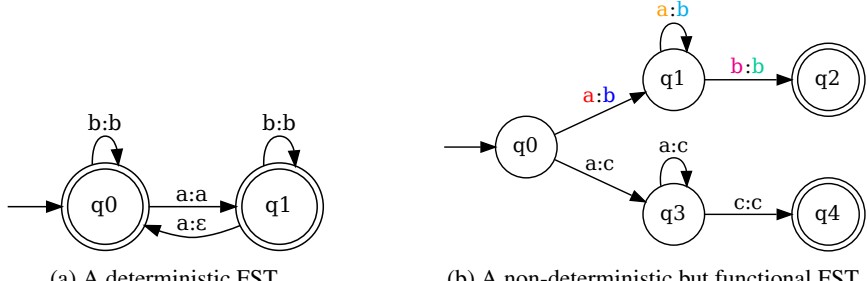

(a) A deterministic FST.  (b) A non-deterministic but functional FST.

Figure 2: Examples of *functional* FSTs. The FST in (a) deletes every other a. The FST in (b) replaces any a in the input string by a b if the last input symbol is a b. Conversely, if the last symbol is a c, any a is replaced by a c. The output can only be determined after the last input symbol.

the input string. In short, while non-deterministic FSTs can take context to the right into account, deterministic FSTs cannot.

## 4 SIMULATION-INDUCED PRIOR

Our approach follows the pre-training and fine-tuning paradigm. We first pre-train on synthetic FST tasks by giving the model a representation of an FST as a prefix and an input string (see Fig. 1). The training objective is to predict the output of the FST on the input string and thereby simulate the behaviour of the FST in the model. Our research hypothesis is that training a model to robustly simulate a broad range of FSTs incentivizes finding reusable mechanisms for FST-like behaviour. When fine-tuning the model using a tunable prefix instead of an encoding of an FST, these mechanisms should be easy to leverage and provide a structural inductive bias for FST-like tasks.

### 4.1 PRE-TRAINING

During pre-training, the model is given a representation of an FST and a string in its domain and has to predict the output of that FST on the given input string. The input to the Transformer is a sequence of vectors from $\mathbb{R}^d$, which consist of a prefix that represents the FST $f$ and a suffix comprised of the embeddings of the input string (see Fig. 1):

$$\underbrace{\mathbf{h}_1, \mathbf{h}_2, \ldots, \mathbf{h}_k}_{\text{FST encoding}}, \underbrace{\mathbf{x}_1, \mathbf{x}_2 \ldots, \mathbf{x}_n}_{\text{Input to FST}}$$

Each $\mathbf{h}$ encodes one transition $p \xrightarrow{\sigma:\rho} q$ of $f$ as a vector:

$$\mathbf{h} = W[\text{EMBED}_{\text{State}}(p); \text{EMBED}_{\text{State}}(q); \text{EMBED}_{\text{Symbol}}(\sigma); \text{EMBED}_{\text{Symbol}}(\rho); \text{EMBED}_{\text{Final}}(e)]$$

where $[;]$ represents vector concatenation, $e$ indicates if $q$ is a final state, and $W$ is linear layer that ensures that $\mathbf{h} \in \mathbb{R}^d$. All embeddings are simple look-up tables based on the id of the state or symbol.[1] The initial state of the FST is always assigned the id 0. Positional embeddings are used as usual. The model is trained to maximize the log probability of the output $y = f(x)$ of the FST $f$.

### 4.2 FINE-TUNING

After pre-training, we can apply our model to a downstream task and fine-tune it. We assume we do not have access to an FST for the downstream task, and therefore we replace the FST encoding with a sequence of tunable embeddings. These embeddings are initialized to the average of the encoding of multiple FSTs from the pre-training phase. The most straightforward way to fine-tune is to only modify the embeddings in the prefix because we are looking for an FST-like representation of the task. This is similar to prompt tuning (Lester et al., 2021). However, this does not work well on tasks outside the pre-training distribution. Therefore, we fine-tune the entire model, including the prefix, and use a higher learning rate for the prefix than for the rest of the model (see Appendix F).

---

[1]This encoding approach neglects that permuting the state numbers has no effect on the function that the FST represents. We leave this to future work, e.g. using graph neural networks.

### 4.3 Constructing Pre-Training Data

To create our pre-training data, we sample 40,000 deterministic FSTs. For every FST, we sample 5 input/output pairs with input lengths up to 35. In total, this leads to 200,000 pairs for training along with their FSTs. To describe the sampling procedure in more detail, we use an overall vocabulary $V$ consisting of the printable ASCII tokens and the Unicode block for IPA symbols (used for transcribing speech). Seq2seq tasks in the wild usually do not use the whole space of this vocabulary, so for each task $T$ we first uniformly sample the vocabulary size $|V_T|$ between 5 and 25 and then uniformly select a subset $V_T \subseteq V$. Then, we uniformly sample the number of states $|Q_T|$ between 2 and 4, and the number of final states between 1 and $|Q_T|$. For every state $q$ and every symbol $\sigma \in V_T$ we introduce at most one outgoing transition to a state $q'$, chosen uniformly at random. This ensures that the FST is deterministic. We then sample the output for the transition: either a symbol $\rho \in V_T$ or $\epsilon$. Finally, we minimize the number of states of the FST using OpenFST (Allauzen et al., 2007), and exclude those without cycles, as they express finite relations. See Appendix A for details.

In practical applications of FSTs, in particular for text editing, one often wants to keep certain parts of the input unchanged. This can be achieved with a set of transitions of the form $q \xrightarrow{\sigma\,:\,\sigma} q'$ for all $\sigma \in V_T$. Since it is very unlikely to sample such a set of transitions, we use a special symbol that acts as a shorthand for this, which we also use when encoding the FST for pre-training.

## 5 Methodology for Measuring Inductive Bias

Inductive biases are the preferences and the abstract knowledge that a learner brings to the task before having seen any data. The inductive bias of a learner helps it fill in the 'gaps' that are not covered by the training data. In order to evaluate inductive bias, we specifically design training data to contain gaps and probe the behaviour of the learner on these gaps. In this paper, we use two different setups that ensure we evaluate on gaps: learning from a small amount of data (few-shot learning) and systematic generalization outside of the training distribution.

We consider a model to have an inductive bias specifically towards FSTs if its behaviour on the gaps in the training data resembles the most plausible FST according to Occam's razor. We consider the FST the most plausible that (i) explains the training data and (ii) all else being equal, is as simple as possible, i.e. has the smallest number of states. [2]

We now describe two methods for constructing data for a given (minimal)[3] FST such that the training and test distributions are different, and that reward a model for inferring the simplest FST.

**Iteration generalization.** A simple form of out-of-distribution generalization is to generalize from short examples to longer examples. In particular, given an FST $f$, we test the ability to generalize from visiting a state only a few times (*iteration count* up to 3) to seeing it more often (iteration count at least 4). A model with an inductive bias for FSTs should be able to obtain high accuracy in this setting. This is because $f$ is the simplest FST that explains the data. Any FST that behaves the same as $f$ on the training data but differs on longer inputs has to have additional states or transitions that were unused on the training data.

**Unseen combinations of transitions.** LSTMs and Transformers struggle to generalize to unseen combinations of known elements (Keysers et al., 2020).

For example, consider the FST $f$ in Fig. 3, which deletes leading zeros from a number. Suppose that a model is trained on examples such as `0012`, `2201`, `1012` but no training example contains the combination of leading zeros followed by a `2` (the combination of the two red adjacent transitions). A model with an inductive bias towards FSTs should nevertheless generalize to this unseen combination and correctly handle examples such as `0021` because $f$ is the simplest FST that explains the data.

---

[2] The preference for simple FSTs is crucial for this to be meaningful. Consider approximating a function $f$ mapping between strings of bounded length with a model $\hat{f}$. Suppose we only required that $\hat{f}$ (i) fit the training data and (ii) correspond to *some* FST. The requirement (ii) is trivially true for any $\hat{f}$, giving any model an inductive bias towards FSTs under this notion. This is because any function between strings of bounded length is defined for finitely many elements, and hence can be represented by an FST.

[3] an FST such that there is no equivalent FST with fewer states

In order to withhold a combination of transitions $\langle t_a, t_b \rangle$, we construct a new FST $f_{\text{train}}$ as follows: We create two copies $f_a, f_b$ of the original FST $f$. In $f_a$, we remove the transition $t_b$; in $f_b$, we remove the transition $t_a$. Then $f_{\text{train}} = f_a \cup f_b$, which can be constructed by introducing a new initial state with $\epsilon$-transitions into the respective initial states of $f_a$ and $f_b$ (right side of Fig. 3). This ensures that any accepting path goes through $f_a$ or $f_b$ but cannot alternate between the two. Hence, $t_a$ or $t_b$ can be used – but not both in the same string. Note that $f_{\text{train}}$ still describes a partial function (rather than a relation) because any accepting path in $f_a$ and any

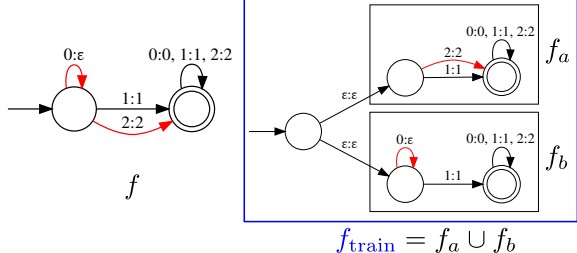

Figure 3: Constructing training data for evaluating unseen combinations of transitions. Based on the given FST $f$, we construct an FST $f_{\text{train}}$ that withholds the *combination* of the two red transitions.

accepting path in $f_b$ is also an accepting path in $f$. As a result, whenever $f_a$ and $f_b$ are both defined, they agree on the result $f_a(x) = f_b(x) = f(x)$. We test exclusively for how a model handles unseen combinations of transitions by generating examples from $f$ for which $f_{\text{train}}$ is *not* defined. We refer to Appendix C for further details.

To make the generalization setup more challenging, these steps can be applied to multiple pairs of adjacent transitions at the same time, i.e. to withhold $\langle t_a^1, t_b^1 \rangle, \ldots, \langle t_a^k, t_b^k \rangle$: We create the copy $f_a$ and remove the transitions $t_b^1, \ldots, t_b^k$ from $f_a$ and analogously remove $t_a^1, \ldots, t_a^k$ from $f_b$.

## 6    EVALUATING SIP'S INDUCTIVE BIAS

In order to understand the effects of our pre-training procedure, we first explore systematic generalization on synthetic FST tasks which allows us to precisely control the similarity between the pre-training and the downstream task.

### 6.1    SETUP AND BASELINES

In order to make a fair comparison, all models we experiment with in the main paper share the same architecture and are initialized from the same checkpoint before any additional pre-training, namely ByT5-small (Xue et al., 2022). This is a Transformer with 300M parameters across 12 encoder layers and 4 decoder layers with a hidden dimensionality of 1472. It was pre-trained on the multilingual C4 corpus. ByT5 uses raw bytes as tokens, which enables full Unicode support and is a natural unit to consider for FST-like tasks such as text editing and grapheme-to-phoneme conversion. We report additional results with a T5-Base model in Appendix E, where we observe similar trends.

**SIP-d4.** This is a model using the method we propose in this work. We pre-train on the data generated in Section 4.3 (**d**eterministic FSTs, with up to **4** states) for 20 epochs. This model achieves an average accuracy of 98% on predicting the output of an unseen FST from the training distribution. For fine-tuning, we use a prefix of length 50 for all experiments in this paper. As an ablation, we also fine-tune the model without the prefix of learnable embeddings (-prefix).

**Naive pre-training.** For this baseline, we use the same pre-training data as for SIP-d4 but we omit the explicit description of the FST and only train on input/output pairs.

**Set.** Wu et al. (2022) investigate the effectiveness of 18 simple synthetic pre-training tasks for a range of downstream tasks, and found Set to perform best on average. The task is to deduplicate characters such that every type occurs only once, e.g. the input `daabacd` becomes `dabc`. This baseline is well-suited for our setup because the task can be represented by a deterministic FST, albeit a very large one with $2^n$ states for a vocabulary of size $n$.

**Task embeddings (TE).** Instead of using an encoding of an FST, this baseline uses 50 randomly initialized embeddings specific to each task (i.e. FST) in the prefix. These embeddings are learned jointly with the model. Several works have used a single token or embedding to encode a domain or task in multi-domain and multi-task learning (Tsvetkov et al., 2016; Stymne et al., 2018; Zhang et al., 2022). Using a shorter tunable prefix resulted in considerably worse performance in our setup.

Table 1: Evaluating systematic generalization on FST tasks with 4 states. We report averages over 5 tasks. ED is edit distance. Due to an outlier task on UC, we additionally report the median after '/'.

| | Iteration | | UC | |
|---|---|---|---|---|
| | Acc↑ | ED↓ | Acc↑ | ED↓ |
| ByT5 | 37.8 | 5.87 | 47.4/57.5 | 1.49/0.93 |
| Naive | 42.6 | 4.41 | 44.9/43.2 | 1.52/1.35 |
| Set | 44.4 | 4.58 | 43.6/42.0 | 1.47/1.31 |
| TE | 61.3 | 2.49 | 57.3/63.1 | 1.13/0.74 |
| SIP-d4 | **94.8** | **0.12** | **73.1/93.3** | **0.61/0.13** |
| -prefix | 84.9 | 0.62 | 61.1/76.3 | 0.99/0.50 |

Table 2: Evaluation on non-deterministic FSTs. We report averages over 5 tasks.

| | Iteration | | UC | |
|---|---|---|---|---|
| | Acc↑ | ED↓ | Acc↑ | ED↓ |
| ByT5 | 83.4 | 0.52 | 83.1 | 0.40 |
| Naive | 83.1 | 0.49 | 84.2 | 0.37 |
| Set | 82.3 | 0.52 | 83.7 | 0.37 |
| TE | 84.2 | 0.49 | 82.7 | 0.42 |
| SIP-d4 | 87.8 | 0.32 | 90.0 | 0.24 |
| SIP-d4+ | 88.2 | 0.30 | 90.5 | 0.22 |
| SIP-nd7 | **89.5** | **0.27** | **91.2** | **0.18** |

## 6.2 SYSTEMATIC GENERALIZATION WITHIN THE PRE-TRAINING DISTRIBUTION

First, we want to establish to what degree the pre-training has conferred any inductive bias on the distribution it was pre-trained on. In particular, we test for systematic generalization to unseen combinations (UC) and higher iteration counts.

**Setup.** For each generalization setup, we generate 5 FSTs with 4 states using the same procedure as for the pre-training, ensuring they have not been seen in the pre-training. To evaluate UC, we withhold the combination of up to 20 pairs of transitions and generate 5000 training examples with lengths 3 to 15 and corresponding test data as described in Section 5. To evaluate iteration generalization, we generate training examples with a maximum iteration count of 3 and test on longer examples of length up to 30 with an iteration count of at least 4. Since the out-of-distribution performance of two checkpoints of the same model can vary significantly, we report averages on the test set of the last 10 epochs.

**Results.** The results can be found in Table 1. On average, SIP-d4 achieves close to perfect accuracy (with one outlier on UC, skewing the mean). TE also shows a clear improvement over the other baselines but SIP-d4 outperforms TE by a large margin. This suggests that SIP-d4 and TE, to a lesser extent, indeed have acquired a stronger inductive bias for FSTs than the other methods. Using SIP-d4 without the tunable prefix leads to a substantial drop in accuracy, highlighting its importance. We analyze the representations learned by SIP-d4 in the tunable prefix in Appendix G.

## 6.3 MORE COMPLEX FSTS

Does the inductive bias introduced by SIP extend beyond the pre-training distribution to more complex FST tasks? To investigate this, we use the same sampling methodology but generate FSTs with more states. SIP-d4 was pre-trained on FSTs with up to 4 states, and we evaluate on FST tasks with 5, 7 and 10 states. Again, we evaluate by measuring out-of-distribution performance for iteration generalization and unseen combinations.

In Fig. 4 we show how the individual models deviate from the accuracy of ByT5 as a function of the number of states in the test FST. We report the absolute accuracies in Table 5 in the appendix. The trends for the two generalization setups are very similar: SIP always performs best by a clear margin regardless of the number of states in the FSTs. As we increase the number of states and move further away from the pre-training distribution, SIP improves less over the baselines. We see a similar pattern for TE but with considerably smaller improvements over ByT5.

## 6.4 NON-DETERMINISTIC FSTS

As shown in the previous section, SIP still works well for more complex FST tasks than seen during pre-training. However, this evaluation focused on the favourable case where both pre-training and evaluation involve the same class of FSTs, namely deterministic FSTs. Deterministic FSTs can only take left context into account (see Section 3), which is a restrictive assumption. Here, we evaluate if the inductive bias conferred by SIP carries over to non-deterministic functional FSTs, i.e. those that can also take context to the *right* into account.

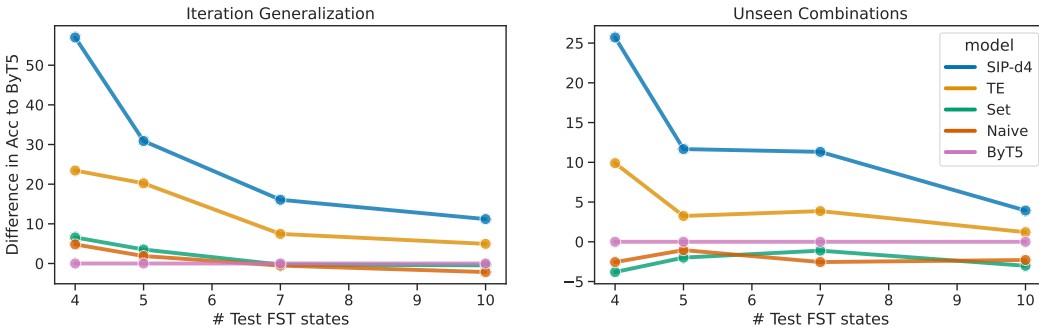

Figure 4: Evaluation on deterministic FST tasks with more states than seen in pre-training. We show the deviation in percentage points from ByT5.

We automatically generate 5 non-deterministic FSTs with 21 states (see Appendix B for details) and report averages in Table 2. Despite the structural mismatch between pre-training and the downstream tasks, SIP-d4 shows clear improvements over the baselines. Interestingly, TE does not consistently outperform the other baselines, despite its stronger results on deterministic FSTs.

Our pre-training procedure does not hinge on using deterministic FSTs. This raises the question if we can achieve even better performance by adjusting the inductive bias. To investigate this, we further pre-train SIP on 40,000 non-deterministic FSTs with up to 7 states, which we call SIP-nd7. To control for the additional training data of SIP-nd7, we also further pre-train SIP-d4 with the same number of deterministic FSTs with the same characteristics as in Section 4.3 (SIP-d4+). The results in Table 2 show better performance of SIP-nd7, which supports the hypothesis that the inductive bias can be adjusted. SIP-d4+ shows a smaller improvement over SIP-d4. Based on 5 additional FSTs per setup to gain more statistical power, we found that the difference between SIP-nd7 and SIP-d4+ is statistically significant ($p = 0.017$, $n = 20$, paired permutation test).

## 7 TRANSFER TO NATURAL DATA

In this section, we investigate to what degree the inductive bias from pre-training on synthetic data transfers to tasks with natural data that have been traditionally approached with finite state methods.

### 7.1 LOW-RESOURCE GRAPHEME-TO-PHONEME CONVERSION

Grapheme-to-phoneme conversion is the task of converting a word as a sequence of symbols (for example, letters in the Latin alphabet) into a description of how this word is pronounced as letters in the IPA alphabet. For example, a possible pronunciation of 'explanation' is [ˌɛkspləˈneɪʃən]. Grapheme-to-phoneme conversion can be part of text-to-speech pipelines and FSTs for this purpose usually are two or three magnitudes larger than the FSTs we constructed for pre-training. Because of this, it enables us to test how far beyond the pre-training distribution SIP remains helpful. We focus on learning from small amounts of data, for which a structural inductive bias towards FSTs should be helpful. We evaluate on 7 low-resource languages from different language families that use their own scripts (Balinese, Coptic, Gothic, Lao, Sylheti, Telugu and Central Atlas Tamazight). We obtained the data from Wikipron (Lee et al., 2020).

As a soft upper bound, we compare with Charsiu (Zhu et al., 2022) which is a ByT5-small model that has been further pre-trained on 7.2 million examples of grapheme-to-phoneme conversion across 100 languages. Although Charsiu was not exposed to the scripts of the languages we chose, it may have seen closely related languages with an overlap in the lexicon from which it can transfer.

The results are in Table 3. The original ByT5-small model performs worst on average despite being a strong model for grapheme-to-phoneme conversion in general (Xue et al., 2022). On average across the languages, SIP-d4 outperforms the other methods that pre-train on synthetic data as well as ByT5. The difference between SIP-d4 and Set is statistically significant ($p = 0.0004$, paired permutation test). On Coptic, SIP-d4 even comes close to Charsiu. Fine-tuning SIP-d4 without the tunable prefix consistently leads to a drop in performance, with the exception of Gothic.

Table 3: Grapheme-to-phoneme conversion with 100 training examples. We show averages of 5 selections of training examples. PER is Phoneme Error Rate: edit distance / length of gold output.

| | ban | | cop | | got | | lao | | syl | | tel | | tzm | | Avg | |
|---|---|---|---|---|---|---|---|---|---|---|---|---|---|---|---|---|
| | Acc | PER | Acc | PER | Acc | PER | Acc | PER | Acc | PER | Acc | PER | Acc | PER | Acc↑ | PER↓ |
| Charsiu | 68.3 | .110 | 7.8 | .579 | 67.0 | .067 | 35.1 | .238 | 47.6 | .196 | 73.3 | .070 | 18.6 | .403 | 45.4 | .238 |
| ByT5 | 50.2 | .233 | 1.0 | .847 | 30.7 | .269 | 1.9 | .760 | 9.8 | .598 | 6.9 | .597 | 2.7 | .851 | 14.8 | .594 |
| Set | 53.9 | .216 | 2.2 | .742 | 58.2 | .094 | 5.8 | .595 | 28.2 | .353 | 27.7 | .293 | 6.4 | .658 | 26.1 | .421 |
| TE | 54.7 | .183 | 1.9 | .756 | 37.0 | .174 | 5.1 | .573 | 30.0 | .309 | 16.2 | .377 | 7.4 | .644 | 21.8 | .431 |
| SIP-d4 | **59.2** | **.152** | **6.6** | **.563** | 56.5 | .096 | **8.2** | **.498** | **39.8** | **.252** | **33.1** | **.228** | **11.0** | **.544** | **30.6** | **.333** |
| -prefix | 55.1 | .168 | 3.2 | .681 | **63.9** | **.072** | 7.8 | .508 | 28.0 | .333 | 28.9 | .252 | 7.0 | .593 | 27.7 | .372 |

Table 4: Averages of accuracy and edit distance across 5-shot text editing tasks based on 8 draws of training examples. We report results grouped by tasks that cannot be solved by a compact FST (reverse-name, surname-initial), tasks that can be solved by FSTs, and overall averages.

| | reverse-name | | surname-initial | | FST | | Overall | |
|---|---|---|---|---|---|---|---|---|
| | Acc↑ | ED↓ | Acc↑ | ED↓ | Acc↑ | ED↓ | Acc↑ | ED↓ |
| ByT5 | 11.8 | 6.81 | 47.2 | 1.76 | 47.6 | 1.42 | 45.7 | 1.72 |
| Charsiu | 43.8 | 1.73 | 52.8 | 0.87 | 62.4 | 0.74 | 60.9 | 0.80 |
| Set | 79.0 | 1.34 | 41.5 | 3.37 | 68.2 | 0.71 | 67.4 | 0.89 |
| TE | 80.3 | 1.08 | 88.2 | 0.41 | **95.7** | **0.11** | **94.5** | 0.17 |
| SIP-d4 | 92.4 | 0.34 | **97.2** | **0.10** | 91.6 | 0.13 | 91.9 | **0.14** |
| -prefix | **97.8** | **0.10** | 72.6 | 0.51 | 89.0 | 0.27 | 91.4 | 0.18 |

## 7.2 FEW-SHOT TEXT EDITING

Learning simple text editing tasks (Jane Doe → J. Doe) from a handful of examples with a Transformer requires a strong structural inductive bias to overcome competing explanations of the data and hence provides a good benchmark for our approach. Text editing has been studied in the context of program synthesis and we evaluate on 19 such tasks from the SyGuS competition 2017 (Alur et al., 2017). Instead of predicting a program, our model directly operates on input/output examples. We note that 17 of these tasks can be solved by compact FSTs, whereas two cannot. These two tasks are *reverse-name* (Jane Doe → Doe Jane) and *surname-initial* (John Doe → Doe, J.), which require tracking information about the first name (either in full or only the initial) in the states.

We report results for 5-shot experiments in Table 4. SIP-d4 and TE excel at this, reaching well above 90% accuracy on average whereas the other methods perform worse by a large margin. Charsiu does not perform clearly better than baselines such as Set – even though it obtains excellent results on grapheme-to-phoneme conversion. Interestingly, TE performs better than SIP-d4 on the tasks that can be solved with FSTs, potentially because the initialization of the prefix for TE follows the same distribution as during pre-training, which is not the case for SIP. However, SIP considerably outperforms TE on the two tasks that cannot be compactly represented by FSTs, suggesting that some of the mechanisms acquired during pre-training can sometimes be leveraged in other contexts as well. In this case fine-tuning SIP-d4 without the tunable prefix leads only to a very small drop in accuracy on average.

## 8 CONCLUSION

We present SIP, a simple and adjustable method for introducing a structural inductive bias into a seq2seq model. Specifically, we focus on an inductive bias towards FSTs, one of the simplest computational device that is useful for NLP applications. We achieve this by pre-training a Transformer to simulate FSTs, i.e. to predict the output of an FST given an input string and a representation of the FST. Our experiments show that our method imparts the desired inductive bias, resulting in improved systematic generalization and better few-shot learning for FST-like tasks. In future work, we plan to extend this methodology to more expressive formalisms such as Pushdown Transducers which can be used for a wider range of downstream NLP tasks.

## REPRODUCIBILITY

We release our code for generating synthetic data and running all experiments as supplementary material, and will put it on github upon publication. The supplementary material also contains the preprocessed natural data needed for reproducing our experiments as well as spreadsheets with raw experimental results, each with the random seed and the configuration (including hyperparameters) that was used. Moreover, we describe additional details about our procedure to generate the deterministic FSTs in Appendix A (including pseudocode), how we generate non-deterministic FSTs in Appendix B, and provide additional information about the model setup and hardware in Appendix F. Upon publication, we will also release our pre-trained model as it is difficult to provide this anonymously.

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

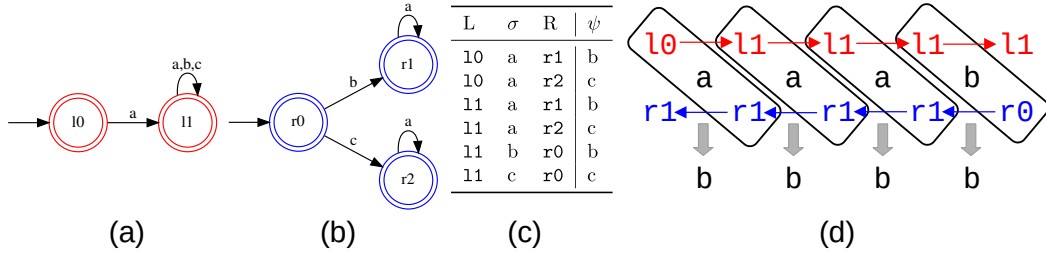

Figure 5: (a) - (c) shows a bimachine that is equivalent to Fig. 2b. (a) Left automaton $A^l$, (b) Right automaton $A^r$, (c) output function $\psi$. (d) shows an example run of the bimachine on the input `aaab` which is mapped to `bbbb`.

Zhuosheng Zhang, Shuohang Wang, Yichong Xu, Yuwei Fang, Wenhao Yu, Yang Liu, Hai Zhao, Chenguang Zhu, and Michael Zeng. Task compass: Scaling multi-task pre-training with task prefix. In *Findings of the Association for Computational Linguistics: EMNLP 2022*, pp. 5671–5685, Abu Dhabi, United Arab Emirates, December 2022. Association for Computational Linguistics. doi: 10.18653/v1/2022.findings-emnlp.416. URL https://aclanthology.org/2022.findings-emnlp.416.

Hao Zheng and Mirella Lapata. Compositional generalization via semantic tagging. In *Findings of the Association for Computational Linguistics: EMNLP 2021*, pp. 1022–1032, Punta Cana, Dominican Republic, November 2021. Association for Computational Linguistics. doi: 10.18653/v1/2021.findings-emnlp.88. URL https://aclanthology.org/2021.findings-emnlp.88.

Jian Zhu, Cong Zhang, and David Jurgens. ByT5 model for massively multilingual grapheme-to-phoneme conversion. In *Proc. Interspeech 2022*, pp. 446–450, 2022. doi: 10.21437/Interspeech.2022-538.

## A GENERATING DETERMINISTIC FSTs

Before describing our procedure for sampling deterministic FSTs, we briefly establish notation. An FST is a tuple $\langle Q, \Sigma, \Gamma, I, F, \Delta \rangle$, where $Q$ is a finite set of states, $\Sigma$ is the input alphabet, $\Gamma$ is the output alphabet, $I \subseteq Q$ is a set of initial states, $F \subseteq Q$ is a set of final states and $\Delta \subseteq Q \times (\Sigma \cup \{\epsilon\}) \times (\Gamma \cup \{\epsilon\}) \times Q$ are the transitions. We assume $\Sigma = \Gamma$ and call it $V$ for vocabulary.

For technical reasons, we exclude the three characters `[`, `]` and `\` from the vocabulary as they are interpreted as special characters by OpenFST, which we use for constructing and representing FSTs.

In addition to the shorthand for identity transitions (`id`), we also have shorthands for converting upper case to lower case and vice-versa (`lower-to-upper`, `upper-to-lower`). We describe our procedure to generate a deterministic FST with pseudocode in Algorithm 1. It receives as argument $n$ (the number of states in the FST), $f$ (number of final states), $V$ (the vocabulary of this FST), and probabilities P-ID, P-DROP, P-SHORTHAND. These probabilities control the likelihood of using a shorthand, not drawing an outgoing edge (P-DROP) with a given symbol, and creating a single identity transition (P-ID). We use CHOICE to denote a uniform random choice from a finite set.

We use P-ID $= 0.2$, P-DROP $= 0.4$, P-SHORTHAND $= 0.15$ in our experiments.

For all experiments with synthetic data, we generate 5000 training examples and 1000 test examples. To reduce variance across tasks, we fix the vocabulary size to its maximum value (25) in the pre-training data and only use printable ASCII characters.

## B GENERATING NON-DETERMINISTIC FUNCTIONAL FSTs

It is not straightforward to directly generate non-deterministic FSTs that are guaranteed to express a function. However, we can directly generate a bimachine, which then can be converted into an FST.

---

**Algorithm 1** Generate a random deterministic FST

---

**function** GEN-DET-FST($n, f, V,$ P-ID, P-DROP, P-SHORTHAND)
    $Q = \{0, \ldots n - 1\}$
    $\Delta = \emptyset$
    $I = \{0\}$
    **for** $q \in Q$ **do**
        $q' = $ CHOICE$(Q)$
        **with prob** P-SHORTHAND
            $s = $ CHOICE$([\texttt{id}, \texttt{lower-to-upper}, \texttt{upper-to-lower}])$
            $\Delta := \Delta \cup \{q \xrightarrow{s\,:\,s} q')\}$
        **else**
            **for** $\sigma \in V$ **do**
                **with prob** P-DROP
                    no-op                           ▷ No outgoing edge with $\sigma$ at $q$
                **else with prob** P-ID
                    $\Delta := \Delta \cup \{q \xrightarrow{\sigma\,:\,\sigma} q'\}$
                **else**
                    $\Delta := \Delta \cup \{q \xrightarrow{\sigma\,:\,\text{CHOICE}(V \cup \{\epsilon\})} q'\}$
                **end with prob**
            **end for**
        **end with prob**
    **end for**
    Eliminate states from $Q$ through which no accepting path can go
    Choose random subset $F$ of $Q$ with $|F| = \min(f, |Q|)$
    **return** minimized FST with states $Q$, transitions $\Delta$, initial states $I$ and final states $F$
**end function**

---

---

**Algorithm 2** Generate output function for bimachine

---

**function** GEN-OUTPUT-$\psi(n^L, n^R, V,$ P-ID $= 0.2)$
    **for** $q^L \in 0, \ldots, n^L - 1$ **do**
        **for** $q^R \in 0, \ldots, n^R - 1$ **do**
            **for** $\sigma \in V$ **do**
                **with prob** P-ID
                    $\psi(q^L, \sigma, q^R) := \sigma$
                **else**
                    $\psi(q^L, \sigma, q^R) := $ CHOICE$(V \cup \{\epsilon\})$
                **end with prob**
            **end for**
        **end for**
    **end for**
    **return** $\psi$
**end function**

---

Bimachines (Schützenberger, 1961) represent exactly the regular string functions, i.e. for every functional FST there is a bimachine that represents it. A bimachine consists of two deterministic finite state automata (called left and right) and an output function. Let $A^L$ be the left FSA with states $Q^L$ and transition function $\delta^L : Q^L \times \Sigma \to Q^L$), and let $A^R$ bet the right FS with states $Q^R$ and transition function $\delta^R : Q^R \times \Sigma \to Q^R$. The output function is $\psi : Q^l \times \Sigma \times Q^r \to \Gamma^*$. All states of $A^L$ and $A^R$ are final states. Given an input string $x = \sigma_1 \sigma_2 \sigma_3 \ldots \sigma_n$, a bimachine runs $A^L$ from left to right over $x$, keeping track of the states $q_0^l, q_1^l, q_2^l, \ldots q_n^l$. It also runs $A^R$ over the string $x$ but this time from right to left, again keeping track of the states $q_0^r, q_1^r, q_2^r, \ldots q_n^r$ that are visited. Then, the state sequence of the right automaton is reversed and $\psi$ is applied 'elementwise' as illustrated in Fig. 5. More formally, the output of the bimachine is $\psi(q_0^l, \sigma_1, q_{n-1}^r)\psi(q_1^l, \sigma_1, q_{n-2}^r)\psi(q_2^l, \sigma_1, q_{n-3}^r) \ldots \psi(q_{n-1}^l, \sigma_1, q_0^r)$.

Table 5: Evaluation on deterministic FSTs with more states, showing absolute accuracies and edit distances, corresponding to Fig. 4.

| Gen. Type | Num States Model | 4 Acc↑ | ED↓ | 5 Acc↑ | ED↓ | 7 Acc↑ | ED↓ | 10 Acc↑ | ED↓ |
|---|---|---|---|---|---|---|---|---|---|
| Iteration | ByT5 | 37.8 | 5.87 | 58.7 | 3.21 | 48.2 | 3.71 | 45.7 | 3.87 |
| | Naive | 42.6 | 4.41 | 60.5 | 2.20 | 47.7 | 3.16 | 43.6 | 3.65 |
| | Set | 44.4 | 4.58 | 62.2 | 2.41 | 48.0 | 3.49 | 45.3 | 3.71 |
| | TE | 61.3 | 2.49 | 78.9 | 0.86 | 55.7 | 2.29 | 50.7 | 2.95 |
| | SIP-d4 | **94.8** | **0.12** | **89.6** | **0.27** | **64.3** | **1.34** | **56.9** | **2.39** |
| UC | ByT5 | 47.4 | 1.49 | 62.6 | 1.05 | 61.9 | 1.29 | 54.1 | 1.70 |
| | Naive | 44.9 | 1.52 | 61.6 | 1.08 | 59.3 | 1.30 | 51.8 | 1.68 |
| | Set | 43.6 | 1.47 | 60.6 | 1.09 | 60.8 | 1.31 | 51.1 | 1.71 |
| | TE | 57.3 | 1.13 | 65.9 | 0.98 | 65.7 | 1.17 | 55.3 | 1.60 |
| | SIP-d4 | **73.1** | **0.61** | **74.3** | **0.69** | **73.2** | **0.85** | **58.0** | **1.44** |

Bimachines can be compiled into FSTs with a simple product construction. For a bimachine $\langle A^L, A^R, \psi \rangle$, one can construct an equivalent FST as follows:

$$\langle Q^L \times Q^R, \Sigma, \Gamma, \{s^L\} \times Q^R, Q^L \times \{s^R\}, \Delta \rangle$$

where $s^L$ and $s^R$ are initial states of $A^L$ and $A^R$, and $\Delta$ contains all transitions

$$\Delta = \{\langle q^L, q^R \rangle \xrightarrow{\sigma : \rho} \langle q'^L, q'^R \rangle \quad | \quad \delta^L(q^L, \sigma) = q'^L, \delta^R(q'^R, \sigma) = q^R, \rho = \psi(q^L, \sigma, q'^R)\}$$

We refer to Mihov & Schulz (2019) for details and further information about bimachines.

In order to sample bimachines, we re-use Algorithm 1 with P-SHORTHAND $= 0$, and ignore the outputs of the transitions, treating them as FSAs. We sample the output function according to Algorithm 2. For the test data creation (Table 2), we use 5 states in the left FSA and 4 states in the right FSA, and set P-DROP $= 0.4$. For creating the training data for SIP-nd7, we use 2 or 3 states in either left or right automaton and set P-DROP $= 0.6$ to keep the length of the prefix low to save GPU memory.

## C UNSEEN COMBINATIONS OF TRANSITIONS

In the main paper, we described how we can withhold combinations of transitions. Here, we briefly describe how we select *which* pairs of transitions we want to withhold. We only select adjacent transitions, i.e. transitions where one can be used immediately after the other. In addition, some transitions cannot be deleted without cutting off a vital initial or final state, which can lead to $f_{\text{train}} = \emptyset$. We ensure this never happens by never withholding the first transition into each state based on a depth-first traversal of the FST.

While this procedure generates an FST $f_{\text{train}}$ that requires more states/transitions than the original $f$, it is unlikely but not *guaranteed* that there is no equivalent FST to $f_{\text{train}}$ that is smaller than $f$.

## D ADDITIONAL RESULTS WITH MORE STATES

In Fig. 4, we show accuracy relative to the accuracy of ByT5. Here, we show the absolute accuracies and edit distances in Table 5.

## E ADDITIONAL RESULTS WITH T5-BASE

We run a subset of the experiments starting off from a pre-trained T5-Base (Raffel et al., 2020) instead of ByT5. This model is about one-third smaller than ByT5 (around 200 million instead of 300 million parameters). T5-Base uses a different vocabulary than ByT5, so we resize the output layer to the vocabulary size of ByT5 and re-initialize it. For the input embeddings, we re-purpose the first $n$ embeddings in the T5-Base embedding matrix to represent the token ids according to

Table 6: Evaluating systematic generalization on FST tasks with 4 states (cf. Table 1). Due to an outlier task on UC, we additionally report the median after '/'.

|  | Iteration | | UC | |
| --- | --- | --- | --- | --- |
|  | Acc↑ | ED↓ | Acc↑ | ED↓ |
| T5-Set | 26.6 | 6.26 | 55.1/54.6 | 1.18/1.02 |
| T5-SIP-d4 | 94.5 | 0.11 | 75.4/99.5 | 0.54/0.01 |

Table 7: Evaluation with T5-Base on non-deterministic FSTs (cf. Table 2)

|  | Iteration | | UC | |
| --- | --- | --- | --- | --- |
|  | Acc↑ | ED↓ | Acc↑ | ED↓ |
| T5-Set | 77.9 | 0.73 | 81.7 | 0.53 |
| T5-SIP-d4 | 83.3 | 0.56 | 86.1 | 0.37 |

Table 8: Grapheme-to-phoneme conversion with 100 training examples based on T5-Base. In contrast to the experiments in the main paper, we found that T5-SIP-d4 did not perform well on completely unseen scripts, so we mapped all Unicode code points to arbitrary ASCII characters. This maintains the structure of the task and is completely reversible. T5-Set is evaluated in the same way.

|  | ban | | cop | | got | | lao | | syl | | tel | | tzm | | Avg | |
| --- | --- | --- | --- | --- | --- | --- | --- | --- | --- | --- | --- | --- | --- | --- | --- | --- |
|  | Acc | PER | Acc | PER | Acc | PER | Acc | PER | Acc | PER | Acc | PER | Acc | PER | Acc↑ | PER↓ |
| T5-Set | 47.9 | .231 | 1.2 | .783 | 6.7 | .458 | 3.6 | .643 | 6.6 | .611 | 4.9 | .612 | 2.7 | .797 | 10.5 | .591 |
| T5-SIP-d4 | 59.1 | .154 | 4.7 | .640 | 69.6 | .059 | 5.9 | .566 | 22.1 | .447 | 35.4 | .191 | 12.5 | .509 | 29.9 | .367 |

the ByT5 tokenizer. While this is suitable as a starting point for further pre-training, we found that directly fine-tuning T5-Base with these modifications led to very poor results and do not include them here. Instead, we train T5-Set (analogous to Set) for a fair point of comparison.

We report a subset of the results from the main paper in for T5-Base in Tables 6 to 8.

## F   ADDITIONAL MODEL DETAILS & HYPERPARAMETERS & HARDWARE

**SIP.** For completeness, we now describe the order in which we arrange the transitions. While the ordering of the transitions does not matter for expressing FSTs, the Transformer uses positional encodings which might have impacts on the pre-training. We assemble the overall prefix by stacking the individual vectors $h_0, \ldots, h_n$ of the transitions $p_0 \xrightarrow{\sigma_0 : \rho_0} q_0, \ldots, p_n \xrightarrow{\sigma_n : \rho_n} q_n$. We group the transitions by their originating state (i.e. $p_i$) and go over the states by their id, starting with 0, the initial state.

During pre-training, we might encounter FSTs with different numbers of transitions within the same batch. To handle this, we use padding encodings by reserving a special padding state and padding symbol in the embedding matrices of states and symbols. To initialize the prefix for fine-tuning, we use the average of 32 FST encodings (chosen at random) from pretraining.

For pre-training, we use embeddings of dimensionality 64 for states, embeddings of dimensionality 256 for symbols, and of dimensionality 16 to indicate final/non-final states.

**Task embeddings.** In order to enable faster adaption of the task embeddings than the rest of the model to fit a particular task, we use a higher learning rate for the task embeddings (1.0) than for the rest of the model ($5 \cdot 10^{-4}$) during pre-training. We also use a higher learning rate for the prefix during fine-tuning, analogously to SIP.

Because we have to store 40,000 task embeddings (one for each generated FST), TE requires a lot of memory. To reduce memory consumption, the task embeddings have a dimensionality of 180 and are up-projected to fit into the Transformer, analogously to W in Section 4.1. Nevertheless, the memory consumption of the embeddings is substantial and we store them on a separate GPU. Analogously to SIP-d4, we pre-train for 20 epochs.

**Naive.** We pre-train for a single epoch only as we found this achieved better results on downstream tasks than training for 20 epochs.

**Set.** We sample 200,000 examples according to the procedure described by Wu et al. (2022) to match our pre-training dataset size. Again, we found it more helpful for downstream task performance to train for a single epoch rather than 20 epochs.

**Fine-tuning Hyperparameters.** The main hyperparameters involved for both SIP and TE are the learning rates for the main model, and (separately) the learning rate of the tunable prefix. We chose these manually. Generally, we found that using a learning rate of 1.0 was a good choice for the prefix. Lester et al. (2021) report a similarly high learning rate to be useful for prompt tuning. For the rest of the model, we found $3 \cdot 10^{-4}$ and $5 \cdot 10^{-4}$ to work well for SIP-d4 and TE, respectively. For few-shot experiments, we use a somewhat smaller learning rate for TE for the main model ($3 \cdot 10^{-4}$). We noticed that T5-SIP-d4 (see Appendix E) was more sensitive to the learning rate choice in general than SIP-d4.

For any experiment, the chosen learning rates can also be found in the spreadsheet with the raw experimental results in the supplementary material.

**Hardware.** We ran our experiments on NVIDIA GeForce RTX 2080 Ti GPUs (11264MiB RAM) with driver version 535.54.03 and cuda version 12.2.

## G   ANALYSIS OF FINE-TUNED PREFIXES

To gain some understanding of how the prefix of tunable embeddings is used by the model and what it contains, we consider the setup of fine-tuning only the prefix and keeping the rest of the model unchanged. That is, all the task-specific information has to be captured in these embeddings. Specifically, we fine-tune on the 5 FSTs from Section 6.2 for iteration generalization for 20 epochs with a learning rate of 0.5.

We explore two questions:

1. Is the model robust towards different permutations of the fine-tuned prefixes? Intuitively, these permutations correspond to changing the order in which transitions are listed, so ideally the model should not be sensitive to that order.

2. Does the fine-tuned prefix represent the task-specific information in a similar way to how FSTs were encoded during pre-training?

To address the first question, we randomly permute the tuned prefixes and compute accuracy on the iteration generalization data before and after permuting the tuned prefixes. We use 20 permutations per learned prefix and average results across the 5 FSTs. Overall, we find that this results only in a small drop in accuracy: the median drop in accuracy is only around 1.3 percentage points, and the arithmetic mean of the drop is around 7.1 percentage points. Most permutations do not have a big impact on how the prefix is interpreted but a few permutations do have a stronger negative impact, skewing the arithmetic mean.

To address the second question, we test if the learned prefix for a task $t$ resembles an encoding of an FST that solves $t$. For each of the 5 FSTs, we generate 10,000 distractors, i.e. FSTs that have the same number of states and use the same vocabulary as the FST solving $t$. We define the similarity of two prefixes $p, q$ as follows:

$$sim(p, q) = \max_{\pi} \frac{1}{n} \sum_i \frac{p_i^T q_{\pi(i)}}{||p_i||_2 \cdot ||q_{\pi(i)}||_2}$$

where $\pi$ is a permutation, and $p_i$ is the $i$-th vector in prefix $p$, and prefixes $p$ and $q$ both have length $n$. That is, we define the similarity between $p$ and $q$ as the highest possible average cosine similarities between positions in $p$ and $q$ that one can achieve by assigning a position in $p$ to exactly one position in $q$ and vice-versa.[4] Taking the maximum over all permutations is justified by our results to the first question above, which showed that the model is largely invariant to different permutations of the tuned prefix.

---

[4]Computing the similarity $sim(p, q)$ is relatively expensive because it involves solving the assignment problem (e.g. with the Hungarian algorithm). Instead of solving the assignment problem exactly, we approximate it with the Sinkhorn algorithm (Sinkhorn, 1964). We then take the output of the algorithm (a matrix of 'soft' assignments) and for each position in $p$, we greedily select a matching position in $q$.

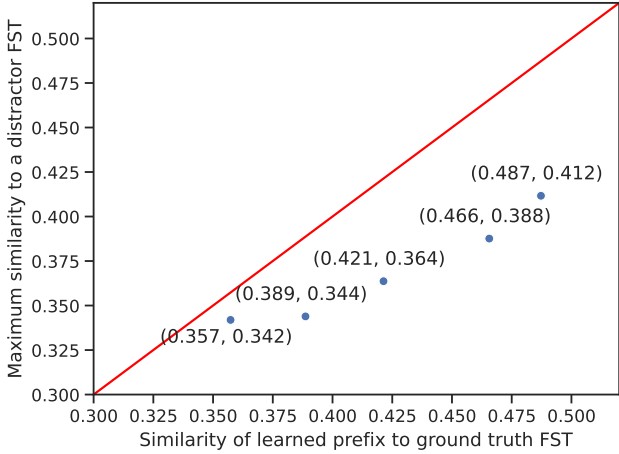

Figure 6: Each dot represents a fine-tuned prefix when the rest of the model remains frozen during fine-tuning. The x-coordinates represent the similarity to a ground truth gold prefix, and the y-coordinates represent the maximum similarity to any of the $5 \times 10000$ distractor FSTs. All dots are below the diagonal, hence all learned prefixes are most similar to an encoding of the ground truth FST.

| Num. states | Split | Min | Max | Mean |
|---|---|---|---|---|
| 4 | train | 2 | 11 | 4.66 |
| 4 | test | 4 | 30 | 18.97 |
| 5 | train | 2 | 14 | 5.39 |
| 5 | test | 4 | 30 | 19.53 |
| 7 | train | 2 | 20 | 6.12 |
| 7 | test | 4 | 30 | 20.13 |
| 10 | train | 2 | 25 | 7.31 |
| 10 | test | 4 | 30 | 20.62 |
| 21 | train | 2 | 30 | 11.80 |
| 21 | test | 5 | 30 | 23.07 |

Table 9: Distribution of input lengths of the train/test data we generate for the iteration generalization experiments in Section 6. The tasks with 21 states are the non-deterministic FSTs from Section 6.4.

For every task $t$, we compute the similarity between the prefix $p$ learned by fine-tuning on input/output pairs and the union of encodings of the distractors and encodings of the gold standard FST for task $t$. Where necessary, we truncate encodings of FSTs to have the same length as the learned prefix. We present the results in Fig. 6 showing that all learned prefixes are most similar to an encoding of the ground truth FST.

## H  LENGTH DISTRIBUTIONS

The input strings in the pre-training data we generate for SIP-d4 have a minimum length of 1, an average length of 15.57 and a maximum length of 35. We report the length distributions for the iteration generalization experiments in Section 6 in Table 9.

## I  GENERALIZATION TO LONGER STRINGS

In the main paper, we report results on iteration generalization where a model is trained on strings such that each state has been visited at most 3 times, and is tested on strings where at least one state is visited at least 4 times. Here, we explore a more extreme version, where there is a large gap between the maximum length seen during training and the minimum length seen during testing. As

Table 10: Average generalization ability across 5 FSTs with 4 states. Models were trained on inputs of length up to 15, and tested on much longer inputs.

| Test length | Model | Max pretrain length | Acc↑ | ED↓ |
|---|---|---|---|---|
| 40 to 70 | ByT5 | 1024 | 29.3 | 15.60 |
| | SIP-d4 | 35 | **69.4** | **2.61** |
| 90 to 110 | ByT5 | 1024 | 1.4 | 55.37 |
| | SIP-d4 | 35 | 3.4 | 34.50 |
| | SIP-d4-long | 110 | **81.5** | **1.09** |

another point of comparison, we further pre-train SIP-d4 on 40,000 FSTs with strings of length up to 110 (SIP-d4-long).

We report results in Table 10. ByT5 struggles with this generalization setup across the board. SIP-d4 performs remarkably well on lengths 40-70 which are beyond the lengths seen during its pre-training. However, performance drops starkly when testing on inputs of length 90 to 110. We hypothesize that this is because the relevant positional embeddings were not pre-trained by SIP. In contrast, SIP-d4-long performs well on inputs of length 90 to 110, as it has seen strings of such length during pre-training.

