# OpenReview forum: "Injecting a Structural Inductive Bias into a Seq2Seq Model by Simulation"
_ICLR.cc/2024/Conference — Submitted to ICLR 2024_

### Official Review · Reviewer_sUW9 · 2023-10-31

**Soundness:** 3 good
**Presentation:** 3 good
**Contribution:** 2 fair
**Rating:** 6
**Confidence:** 3

**Summary:**

The paper explores the direction of injecting inductive biases (into Transformers) by modifying the data (particularly, through synthetic pre-training). The paper tries to inject Finite State Transducer (FST)-like biases by generating relevant synthetic pre-training data. The paper pre-trains a Transformer on the synthetic data and test it for OOD generalization during fine-tuning different FST tasks. The paper found better OOD generalization with FST pre-training and prefixes than other baselines. The paper also shows positive transfer on some natural language tasks.

**Strengths:**

1. Decent focused exploration on injection of inductive bias through synthetic pre-training.

2. Shows the ability to demonstrate OOD generalizations (iteration generalization and systematic generalization) in FST-tasks from synthetic pre-training.

3. Shows transfer from pre-training on FST to some specific natural language tasks.

**Weaknesses:**

1. If By-T5 is already pre-trained in natural data before the synthetic pre-training, I wonder how much of an influence there is from the "pre-pre-training" in enabling OOD generalization and such.

2. The scope feels limited. We already have prior works showing the viability of synthetic pre-training and knowledge transfer from natural language tasks. It is not as clear what the motivation for exploring particularly FST-related tasks is. Transformers have been shown to underperform in OOD generalization on logical inference [1,2], ListOps [2], Flip-Flop languages [3], parity tasks/sensitive tasks [4], automata tasks [5], and others [6]. It would have been good to contrast the approach with some of such works, reconcile with them, and see if the synthetic pre-training proposed here can be used.

[1] The Importance of Being Recurrent for Modeling Hierarchical Structure - Tran. et al. EMNLP 2018

[2] Ordered Memory - Shen et al. NeurIPS 2019

[3] Exposing Attention Glitches with Flip-Flop Language Modeling - Liu et al. NeurIPS 2023

[4] Simplicity Bias in Transformers and their Ability to Learn Sparse Boolean Functions - Bhattamishra et al. ACL 2023

[5] Transformers Learn Shortcuts to Automata  - Liu et al. ICLR 2023

[6] Neural Networks and the Chomsky Hierarchy - Delétang et al. ICLR 2023

**Questions:**

1. What are the average and maximum sequence lengths in the pre-training data, training data, and iteration generalization data?

2. Would it be possible to explore generalizations to higher lengths e.g. 100 or more?

---

> ### Author Response · Authors · 2023-11-17
> **Thank you for your review!**
>
> Thank you for your review!
>
> > If By-T5 is already pre-trained in natural data before the synthetic pre-training, I wonder how much of an influence there is from the "pre-pre-training" in enabling OOD generalization and such.
>
> We tried to train a ByT5-style model from scratch. However, none of the hyperparameter configurations we tried resulted in a model that would converge well. We hypothesize that a model has to be in a reasonable space to begin with to make learning easier.
>
> While we cannot exactly attribute what part of the generalization ability comes from the _combination_ of the original ByT5 weights and our pre-training, ByT5 on its own does not have a strong inductive bias for FST-like tasks and performs worse in our experiments.
>
> > We already have prior works showing the viability of synthetic pre-training and knowledge transfer from natural language tasks.
>
> We briefly review some methods for pre-training with synthetic data in our Related Work section. In contrast to these, our methodology relies on simulating a computational device (FSTs, specifically).
>
> We view our contribution not only as a pretraining procedure but it also includes a specific fine-tuning method (prefix of tunable embeddings) mirroring the pre-training. We have now included an ablation of the tunable prefix during fine-tuning (see Tables 1,3,4). The results show that using a tunable prefix is important for achieving the best performance.
>
> > The scope feels limited [...] It is not as clear what the motivation for exploring particularly FST-related tasks is.
>
> FSTs are an important computational device for NLP on the level of phonology and morphology. Our methodology of simulating a computational device is also quite general, and we view this as a stepping stone to more expressive computational devices with broader applications, such as Pushdown Transducers which could help with processing hierarchical structure. We hope that the results of this work can inform that exploration.
>
> > What are the average and maximum sequence lengths in the pre-training data, training data, and iteration generalization data?
>
> In the pretraining data: the average input sequence length is 15.57 and the maximum is 35. The distribution of the training/test length for iteration generalization depends on the number of states in the FST:
> |  Num States | Data  | Min | Max | Average |
> | ----------- | ----- | --- | --- | ------- |
> | 4           | train | 2   | 11  | 4.66    |
> | 4           | test  | 4   | 30  | 18.97   |
> | 5           | train | 2   | 14  | 5.39    |
> | 5           | test  | 4   | 30  | 19.53   |
> | 7           | train | 2   | 20  | 6.12    |
> | 7           | test  | 4   | 30  | 20.13   |
> | 10          | train | 2   | 25  | 7.31    |
> | 10          | test  | 4   | 30  | 20.62   |
>
> > Would it be possible to explore generalizations to higher lengths e.g. 100 or more?
>
> We ran an experiment on FSTs with 4 states (i.e. from the pre-training distribution), where models are trained on inputs of length up to 15, and tested on inputs of length 40 - 70, and on lengths 90 to 110. Note that this is beyond the lengths seen during pre-training. The results are below (averaged over 5 tasks), and show that our pretraining still has a strong positive impact:
>
>
> |  Model      | Max pre-training length | Test length | Acc  | Edit distance |
> | ----------- | ----------------------- | ----------- | ---- | ------------- |
> | ByT5        | 1024                    | 40 to 70    | 29.3 | 15.6          |
> | SIP-d4      | 35                      | 40 to 70    | **69.4** | **2.61**          |
> | ByT5        | 1024                    | 90 to 110   | 1.4  | 55.37         |
> | SIP-d4      | 35                      | 90 to 110   | 3.4  | 34.5          |
> | SIP-d4-long | 110                     | 90 to 110   | **81.5** | **1.09**          |
>
> We hypothesize that the accuracy of SIP-d4 drops on lengths 90 to 110 because the relevant positional embeddings were not pre-trained.
> SIP-d4-long is a version of SIP-d4 that has been further pre-trained on strings of length up to 110. We refer to the new Appendix I for more details.

---

> > ### Comment · Reviewer_sUW9 · 2023-11-17
> >
> > Thank you for the additional feedback and data. They should improve the paper.
> >
> > > However, none of the hyperparameter configurations we tried resulted in a model that would converge well. We hypothesize that a model has to be in a reasonable space to begin with to make learning easier.
> >
> > This is interesting and could be highlighted in the paper.

---

> > > ### Author Response · Authors · 2023-11-21
> > >
> > > Thank you!
> > >
> > > > This is interesting and could be highlighted in the paper.
> > >
> > > We will mention this.

---

### Official Review · Reviewer_Tami · 2023-11-02

**Soundness:** 3 good
**Presentation:** 3 good
**Contribution:** 3 good
**Rating:** 6
**Confidence:** 3

**Summary:**

In this paper, the authors inject an inductive bias towards Finite State Transducers (FSTs) into a Transformer by pre-training it to simulate FSTs given their descriptions. The proposed method is simple, adjustable and efficient to inject a structural inductive bias for FST-like
tasks into a Transformer. Experimental resuts show that the proposed method has better systematic generalization on tasks beyond the pre-training distribution and strong results when transferring to natural FST-like data, as demonstrated on low-resource grapheme-to-morpheme conversion.

**Strengths:**

1. The proposed method, which involves injecting Finite State Transducers (FSTs) into a Transformer, is novel and presents a promising direction for solving complex tasks in the real world.

2. This paper is well-written and the method is clearly described.

3. Experimental results show that the proposed method can outperform previous work in a wide range of tasks.

**Weaknesses:**

1. The author needs to provide more ablation experiments and analysis to understand the changes brought about by the FST on the results and why it can bring consistent improvements.

2. Despite the many experiments conducted in this paper, I still hope that the authors can apply the method to large language models. It is very important to determine whether the proposed method is still effective in LLMs and whether it can solve some problems existing in larger models, such as hallucinations.

**Questions:**

n/a

---

> ### Author Response · Authors · 2023-11-17
> **Thank you for your review!**
>
> Thank you for your review!
>
> > [...] more ablation experiments and analysis to understand the changes brought about by the FST on the results and why it can bring consistent improvements.
>
> We have included an analysis in Appendix G that gives some insights into what the learned prefix can represent. In a nutshell, when finetuning on tasks from the pre-training distribution and adapting only the prefix (i.e. keeping the rest of the model frozen), the learned prefix resembles an encoding of the ground truth FST.
>
> We have also included an ablation where we don’t use a prefix of tunable embeddings during fine-tuning (see Tables 1,3,4). The results show that a tuneable prefix is important for achieving the best performance.
>
> > Despite the many experiments conducted in this paper, I still hope that the authors can apply the method to large language models. It is very important to determine whether the proposed method is still effective in LLMs and whether it can solve some problems existing in larger models, such as hallucinations.
>
> In future work, we are planning to integrate pre-training to simulate computational devices (e.g. FST or grammars) into LLMs. This will likely require training on a mixture of simulated data and other pre-training data, so will be computationally expensive.
>
> We think that simulating computational devices could potentially help with some problems of LLMs, in particular systematic generalization, but also specific forms of hallucinations, where the model ignores a rule and resorts to default behaviour (see also the [MemoTrap dataset](https://github.com/liujch1998/memo-trap)). For example, consider the following simple text editing problem (similar to Section 7.2)
> |Input | Output|
> | ------ | ------- |
> | Howard Phillips Lovecraft | H.P. Lovecraft |
> | John Ronald Reuel Tolkien | J.R.R. Tolkien |
> | Thomas Stearns Eliot | T.S. Eliot |
>
> The current version of ChatGPT outputs “J.K. Rowling” for the name “John Edward Rowling”, hallucinating the K.

---

> ### Author Response · Authors · 2023-11-21
>
> Please feel free to reach out if you have any additional questions or would like any clarification. We'll do our best to respond in the remaining 40 hours before the discussion period ends.

---

### Official Review · Reviewer_Lr31 · 2023-11-08

**Soundness:** 3 good
**Presentation:** 3 good
**Contribution:** 3 good
**Rating:** 6
**Confidence:** 2

**Summary:**

This paper creates an approach (SIP) to effectively combine FST’s topology information into transformer, through pre-training on vast amount of sampled FSTs and fine-tuned on a tunable embeddings. The author demonstrated various experimental results to support their claim that SIP is able to inject the inductive bias from FSTs into downstream tasks to achieve better performances.

**Strengths:**

Overall, I think the approach of injecting inductive bias that is end2end trainable is interesting, though limiting the capacity of FST by only allowing deterministic designs. The approach treats FST as a prompt prefix / soft-prompt, which is shown to help generation on “FST-like” tasks and allow the incorporate of FSTs within a larger transformer-based neural model.

1.Inductive bias from FST topology as soft-prompt for transformer supports end2end training

2.The idea of simulation prior for generalization is interesting and the author proposes an interesting pipeline (FST data synthesis for pre-training then use average encoding for downstream task). Though I wonder what exactly is learned from the use of turnable encoding for downstream task.

**Weaknesses:**

I don’t think it’s very surprising that transformer is able to learn complex structure encoded from FST, especially when the pre-training data is synthetically generated with a small amount of states.
Moreover,  the use of the current proposed framework would be limited when #states/transition explodes, in addition to that fact that positional embedding from transformer is used as normal, which made encoding of identical FSTs with different state ordering represent different things. The need to encode FST as a sequence of prefix encoding also makes the design of FST topology limited. Overall, I feel this work might scarifies too much symbolic information obtainable from FSTs in order to fit it into the prefix encoding framework.

1.Limitation of FST which has to be deterministic.

2.The setup looks up state and transition from the embedding table which is not scalable.

3.Positional information of transformer is used as usual, which means two identical FST with different state ordering have different representation in the transformer. This could result in unwanted behavior.

**Questions:**

1.What exactly is learned from the tunable embedding? As the author assumes no FST information is available from downstream task, such embedding should be tuned toward certain encoding from pre-training stage right? Have the authors performed any analysis on such embedding?

2.The nature of FST accepts compositional design (that is, addition or composition of different FSTs can be easily combined). I wonder if the proposed approach, when trained on different FSTs encoding, would generalize well to the task that is solvable by their composition?

---

> ### Author Response · Authors · 2023-11-17
> **Thank you for your review!**
>
> Thank you for your review!
>
> > 1. Limitation of FST which has to be deterministic. / The need to encode FST as a sequence of prefix encoding also makes the design of FST topology limited
>
> Any FST can be encoded with our method (as you mention though, the encoding is not unique, e.g. because of different orderings of transitions, see also our reply to your third point). In addition to our experiments with deterministic FSTs, we pretrain on non-deterministic FSTs in section 6.4, which improves OOD accuracy on tasks with underlying non-deterministic FSTs in comparison to pretraining on deterministic FSTs.
>
> Our framework could also easily be extended to stochastic/weighted FSTs by including the weights in the prefix as well.
>
> > 2.The setup looks up state and transition from the embedding table which is not scalable
>
> > the current proposed framework would be limited when #states/transition explodes
>
> While encoding very large FSTs into a prefix can make pre-training expensive (due to the quadratic runtime complexity of the transformer), our results in section 6.3 show that improvements can already be achieved by pre-training on smaller FSTs, which mitigates this issue to a certain extent.
>
> We would also like to mention that other works in this area use black box meta-learning, where the prefix contains a small synthetic dataset of input/output pairs [1, 2]. Instead of encoding a synthetic dataset, we encode the underlying rules of the task, which is much more scalable.
>
> > 3.Positional information of transformer is used as usual, which means two identical FST with different state ordering have different representation in the transformer. This could result in unwanted behavior.
>
> We ran an experiment to test how sensitive our pre-trained model is to different orderings of the transitions in the prefix: when given an encoding of an FST (from the same distribution as the training data), and a version of the encoding where the order of transitions is randomly permuted, the model predicts the same output in 98.3% of the cases. In the cases where the predictions are not exactly the same, they differ by 1.65 characters on average. We hypothesize that the transformer learns to encode them in a relatively position-agnostic way (e.g., by suppressing positional encodings); however, we did not perform a 'mechanistic' analysis of how it has been achieved.
>
> To a lesser extent, this robustness towards permuting the prefix holds also in the fine-tuned case, when only the prefix is tuned and the other model parameters are frozen during finetuning: randomly permuting the order of the vectors in the tuned prefix reduces accuracy by 7.0 percentage points on average, with a median reduction of only 1.3 points (see the new Appendix G for details).
>
> > 1.What exactly is learned from the tunable embedding? [...] such embedding should be tuned toward certain encoding from pre-training stage right?
>
> We performed an analysis that computes the cosine similarity between the tuned prefix of a model (when only the prefix is tuned and the rest of the model is frozen) and an encoding of the “ground truth” FST. This analysis revealed that the learned prefix is reliably more similar to the ground truth FST than to 10,000 different FSTs with similar properties (see the new Appendix G for details). This shows that information is represented in the prefix in a somewhat similar way to encodings of symbolic FSTs.
>
> > 2. The nature of FST accepts compositional design (that is, addition or composition of different FSTs can be easily combined). I wonder if the proposed approach, when trained on different FSTs encoding, would generalize well to the task that is solvable by their composition?
>
> Thank you for suggesting this interesting research direction! It is not clear what the composition function could be for the encoding learned through fine-tuning. However, we think it might be possible to adapt the model to the composition of two FSTs when their encodings are available using a small amount of training data from the composition of the FSTs. There may also be a way to meta-learn the model to accept a certain composition function (e.g., involving the concatenation of prefixes).
>
>
> [1] Compositional generalization through meta sequence-to-sequence learning - Lake. NeurIPS 2019.
>
> [2] Human-like systematic generalization through a meta-learning neural network - Lake and Baroni, Nature 2023.

---

> ### Author Response · Authors · 2023-11-21
>
> Please feel free to reach out if you have any additional questions or would like any clarification. We'll do our best to respond in the remaining 40 hours before the discussion period ends.

---

> > ### Comment · Reviewer_Lr31 · 2023-11-22
> >
> > Hi, thanks for your response, and sorry for the delay!
> >
> > I've read your updated version carefully, and I believe it resolves most of my concerns. Still, I think the FST setup in this paper is not sophisticated enough, maybe adding more complex FSTs (like stochastic/weighted FSTs mentioned by the authors) may help.
> >
> > Besides, I have an open question for the authors: SIP mainly focuses on Grapheme-to-phoneme tasks, which are relatively easy to solve and may suffer from limited practical impact (I understand most FST research does not play a chase game for practicability but for fun). Any ideas to extend SIP to more complex language reasoning tasks?
> >
> > Overall, I increased my score to 6.

---

> > > ### Author Response · Authors · 2023-11-22
> > >
> > > Thank you for your response and for raising the score!
> > >
> > > > SIP mainly focuses on Grapheme-to-phoneme tasks, which are relatively easy to solve and may suffer from limited practical impact (I understand most FST research does not play a chase game for practicability but for fun). Any ideas to extend SIP to more complex language reasoning tasks?
> > >
> > > Few-shot text editing has practical applications, e.g. for cleaning and pre-processing data. Doing this in a data-driven few-shot manner is particularly relevant to users who don’t know how to program, see Microsoft Excel's [Flash Fill](https://support.microsoft.com/en-us/office/using-flash-fill-in-excel-3f9bcf1e-db93-4890-94a0-1578341f73f7), but can also be convenient for data scientists.
> > >
> > > Regarding task complexity, we are currently exploring applying SIP to tasks such as semantic parsing (i.e. translating natural language utterances to database queries) and translating between programming languages. Instead of simulating FSTs for pre-training, we are looking into more expressive computational devices (pushdown transducers, synchronous grammars).

---

### Author Response · Authors · 2023-11-17
**Summary of changes**

Thank you to all the reviewers for your thoughtful reviews and questions! We have updated the paper, and here is the summary of the additions:

- Analysis of what the tunable prefix represents after fine-tuning, which shows that it tends to be more similar to embeddings of ground-truth FSTs than to random FSTs with similar properties (Appendix G)
- Analysis showing the robustness towards permuting the order in the fine-tuned prefix, showing that the model approximately captures that different orders are equivalent  (Appendix G)
- Ablation of the tunable prefix during fine-tuning (see Tables 1,3,4 and the parts of the main paper highlighted in blue), highlighting the importance of the tunable prefix for achieving the best performance.
- Experiments on generalizing to much longer inputs than seen during fine-tuning (Appendix I)
- Statistics about the distribution of lengths in the pre-training data and in the iteration generalization setup (Appendix H)

---

### Meta-Review · Area_Chair_uQBR · 2023-12-06

**Metareview:**

This paper proposes to imbue FST-like inductive biases to Transformers via pretraining the Transformer on data generated from an FST. This approach is found improve performance on synthetic (new FST) and real (grapheme-to-phoneme, few-shot text editing) benchmarks.

Overall, I liked the paper quite a bit. The idea of using a structured model to generate data with which to regularize a Transformer is interesting, and although pretraining on synthetic data has been explored before in various settings (as discussed extensively in the paper), it has not been done with FSTs to my knowledge. However, this paper's focus on FSTs, and applications to (too) simple real-world benchmarks, prevent me from recommending this paper for acceptance. Reviewer sUW9 summarized my concerns well:

> The scope feels limited. We already have prior works showing the viability of synthetic pre-training and knowledge transfer from natural language tasks. It is not as clear what the motivation for exploring particularly FST-related tasks is. Transformers have been shown to underperform in OOD generalization on logical inference [1,2], ListOps [2], Flip-Flop languages [3], parity tasks/sensitive tasks [4], automata tasks [5], and others [6]. It would have been good to contrast the approach with some of such works, reconcile with them, and see if the synthetic pre-training proposed here can be used.

There are also many interesting questions that were left unexplored: for example, what if you condition on an FST that you induce from actual data? What about extensions to other transducers that might better capture the types of transformations required for more complex tasks (e.g., machine translation)? I believe a more thorough exploration of this space would result in a very interesting future submission.

**Justification For Why Not Higher Score:**

Limited scope.

**Justification For Why Not Lower Score:**

N/A

---

### Decision · Program_Chairs · 2024-01-16

Reject